# "It is what we have been told to do": Masculinities and femininities crossing with sexual orientation and feminist activism in Spain

**Ariadna Cerdán-Torregrosa**[1]*, **Daniel La Parra-Casado**[2], **Carmen Vives-Cases**[1,3]

**1** Department of Community Nursing, Preventive Medicine and Public Health and History of Science, University of Alicante, Alicante, Spain, **2** Department of Sociology II, University of Alicante, Alicante, Spain, **3** CIBER of Epidemiology and Public Health, CIBERESP, Madrid, Spain

* ariadna.cerdan@ua.es

## Abstract

Masculinities and femininities are often characterized by social inequalities and mainly studied from the perspectives of adult, heterosexual and non-activist people. This study explores the discourses on masculinities and femininities of young cisgender men and women, involved or not in feminist activism and of different sexual orientations (heterosexual, bisexual and homosexual) in Spain. Between 2019 and 2020, we conducted a qualitative study with 20 semi-structured interviews and 8 discussion groups in which 73 people participated. A socially dominant gender discourse was identified, which establishes a dichotomous understanding of masculinity in constant confrontation with femininity, as well as of heterosexuality against homosexuality, in a heteronormative context. Young people discursively position themselves differently from said discourse according to their sexual orientation and involvement in feminist activism. We encounter discursive positions that reproduce the socially dominant gender discourse, especially detected among non-activist heterosexuals. We also observe others that try to transgress it, mainly among activists of all sexual orientations and non-activist homosexuals and bisexuals. This study empirically adds to the knowledge of the configuration of inequalities in gender relations and how interactions with sexuality take place. The results also provide guidance for future gender-transformative interventions to promote gender equality and social justice.

## Introduction

Gender can be understood as an array of symbolic, affective, socio-economic and power relationships that shape social practices and the way in which men and women interact on an intrapersonal, interpersonal and institutional level in a given culture [1]. From a constructionist perspective, a large part of the ways men and women behave and think are influenced by the concepts of femininity and masculinity that they internalize from their culture [2]. Social gender roles, categorized as masculine or feminine, have been included in people's lives since

Alicante in Spain (reference number UA-2019-04-15) for researchers who meet the criteria for access to confidential data. Excerpts relevant to the study are available within the paper. Access to additional information may be possible under request to the project principal investigator (contact: carmen.vives@ua.es).

**Funding:** This work was supported by GENDER NET Plus Co-Fund (Reference 2018-00968) and the Ministry of Science and Innovation of Spain (Reference PCI2019-103580) as part of the PositivMasc project from which Carmen Vives-Cases is the principal investigator. This study was also conducted within the predoctoral grant received from the Ministry of Universities of Spain (Reference FPU19/00905) from which Ariadna Cerdán-Torregrosa is recipient. The publication costs were supported by AICO, Generalitat Valenciana (2022-2024) (Reference CIAICO/2021/019). The funders had no role in study design, data collection and analysis, decision to publish, or preparation of the manuscript. There was no additional external funding received for this study."

**Competing interests:** The authors have declared that no competing interests exist.

early socialization [3] and they are often characterized by inequalities between and within genders [4].

Connell's theory of "hegemonic masculinity" has been essential for studies on masculinities, as well as influencing the interdisciplinary understanding of gender in general [5]. Hegemonic masculinity is defined as a form of masculinity that legitimizes unequal gender relations between men and women, between masculinities and femininities, and among masculinities. In Western societies, hegemonic masculinity defines "real men" as physically strong, invulnerable, competitive, independent and powerful, among other features [1]. This hegemonic configuration is constructed in relation to other non-hegemonic forms: complicit, subordinate and marginalized masculinities. Connell also establishes the concept of "emphasized femininity", which involves the complementary, compliant and accommodating subordinate relationship of alliance with hegemonic masculinity [5]. Alongside these ideas, Howson refers to two forms of femininity: "ambivalent femininities" and "protest femininities", such as those who have the initiative to transgress traditional gender constructions [6]. In this empirical article, we unify this literature in order to contribute to understanding the interaction between masculinities and femininities.

Gender relations have been structured through power imbalances, precisely the domination that connects hegemonic masculinity asymmetrically to all the various subaltern masculinities and femininities [5, 6]. In addition to the aforementioned, hegemonic masculinity and emphasized femininity relationships underpin heteronormativity as a foundational organizing principle in Western cultures [7]. Heteronormativity is established under a discursive structure in which only two naturally opposite and complementary sexes are considered (male and female), where gender is interpreted as an expression associated with sex (masculinity with male and femininity with female) and is also conceived to be natural for the opposite sexes to attract each other (heterosexuality) [8, 9].

Following the theory of becoming a subject (gendered and sexual), people have been socially subjected to an imperative discourse on how to understand gender and sexuality [10, 11]. Hegemony has not only determined the relations of dominance and subordination between men and women in cultural, organizational, legal and interpersonal practices, but it has also placed people who are not heterosexual in a vulnerable and subordinate position [4]. People are simultaneously subject to and are subjects of this framework [12] through which they continually debate between constriction and possibility [10]. It is essential to delve into the interactions between gender and sexuality in order to attend to the mechanisms by which a part of the population is subordinated and, in this way, contribute to a social change towards equity.

Perceived gender roles are socially constructed standards, they are not static, therefore gender hierarchies are subject to change depending on the social and historical context [1]. Research with young people highlights that there are hybridizations and contradictions in their conceptualizations of femininity between the idea of "new girl", free from the confinements of emphasized femininity, versus the idea of a "fragile girl", who is vulnerable and voiceless [13, 14]. Other studies on masculinities among the young population also point to the emergence of "hybrid masculinities" [15] that include progressive elements while continuing to perpetuate unequal gender relations through more nuanced forms [e.g., 16, 17].

In Spain, gender relations have historically been shaped by several contextual factors. During the Franco regime, a series of laws and practices based on a strong national Catholic ideology resulted in unequal socio-political positions for men and women. As in other countries such as Ireland, Sweden or Israel [18], Spanish men were accorded political and civil rights and educated in a strong military and patriotic context, where masculinity meant authority and decision-making over all family members. Women were assigned the role of mothers and

wives, being educated to be able to care for children, men and the home [19, 20]. After the death of Franco in 1975, Spain's current Constitution was enacted in 1978, which included gender justice. This was reflected in Organic Law 1/2004 of 28th December, on integrated protection measures against gender-based violence, as well as the approval of same-sex marriage in 2005.

The Spanish feminist strike on 8th March 2018 was an example of the strength of feminist activism these days, as it led to strikes with a large popular following supported by mainstream trade unions proposing the cessation of women's paid and unpaid work on that day [21, 22]. Feminist movements in Spain have achieved important policy changes, such as the recent legislation about sexual violence enacted in 2021 that integrates strategies to address emerging forms of violence against women. However, there is no current legislation that addresses the role of masculinities and femininities with gender-based violence. Young people involved in feminist activism occupy a key role, as their gender consciousness-raising efforts and aspirations towards feminist ideals could be translated into political action. Available studies with young activists in other countries such as Ecuador and Peru [23, 24] identify in their narratives the pretension to change cultural discourses and practices about gender in the family, household and intimate partnerships, providing alternatives through social advocacy. Until now, research conducted in Spain has mainly focused on studies on masculinities based on the perspectives of non-activist heterosexual men [e.g., 25, 26]. It is of interest to analyze how gender hierarchies are understood within and through their feminist activism mobilization, as they may shed light on disruptive discourses that might differ from those of non-activists.

There are still social problems in Spain that point to the fact that equality has not been achieved yet. In recent years, concern for the rise in the prevalence of gender-based violence among young people has increased [27]. According to the results of the representative macro survey carried out by the Spanish Government Office against Gender Violence in 2019, 1 in every 2 women over 16 years (57.3 percent) living in Spain has suffered violence (sexual, physical, harassment or stalking by partners or people who have not been partners) throughout their lives, and young people between 16 and 24 years old declare it to a greater extent [28]. In addition, despite recent data indicating that Spain is one of the countries with the highest level of acceptance of homosexuality [29], other research warns that there is still a high prevalence rate of LGBT-phobic bullying [30, 31].

The aim of this study is to explore the discourses on masculinities and femininities of young cisgender people aged between 18 and 24 years, involved or not in feminist activism and of different sexual orientations (heterosexual, bisexual and homosexual) in Spain. Therefore, we pay closer attention to the interplay of femininities and masculinities, and how sexual orientation and involvement or not in feminist activism may influence the way in which femininity and masculinity are constructed by the Spanish youth.

## Methods

### Design

Data were collected using discussion groups and semi-structured interviews. The most suitable techniques were assessed to delve into the meanings and the intersubjective points of view among the participants. Discussion groups were relevant to examine how masculinities and femininities are collectively constructed as these groups are characterized for being flexible, open and little directive [32]. One on one semi-structured interviews enabled more intimate and private spaces to address the topic of study [33].

This study was part of the wider ongoing EU research project PositivMasc [34] whose objective is to identify the discourses that young people and stakeholders use to conceptualize

masculinities and violence against woman (VAW) and the strategies needed to promote anti-VAW masculinities.

## Sampling and participants

The study included young people between 18 and 24 years of age. The reason behind selecting this range of age is not only due to the potential when exploring emerging social behaviors (related to the cohort effect), but also because it comprises a liminal stage when gender identities and sexual orientation are usually in a process of interpretation, search and settlement (related to the life cycle effect).

The structural criteria applied to select the participants were: age (between 18 and 24), gender identity (cisgender men and women), sexual orientation (heterosexual, homosexual and bisexual) and involvement or not in feminist activism. We focused on these criteria in order to achieve structural representativeness within these categories and not to saturate the analytical combinations. Those who participated as feminist activists were actively involved in a feminist non-profit organization or association when the fieldwork of this study was conducted.

In this same sense, the fieldwork was conducted in two contrasting geographical settings: the province of Alicante and the city of Madrid, Spain. Both locations concentrate different population sizes and lifestyles, therefore providing a greater variety in the sociodemographic profiles of the young people.

The applied sampling was of an intentional non-probabilistic type, diversifying with different contactors and starting points in order to guarantee maximum diversity. Some contactors were in charge of finding people for the discussion groups and others were in charge of finding people for the semi-structured interviews. The research team began sampling in two ways: acting themselves as contactors and searching for people from different areas, such as educational institutions (high schools, universities, etc.), non-governmental organizations (NGOs) and state organizations that work with young people. Other strategies used to recruit participants were through flyers at the aforementioned places or online advertisements using social media platforms (Facebook, Instagram, etc.).

A total of 73 young people took part in the study (39 men and 34 women) as observed in Table 1. We conducted a total of 20 semi-structured interviews (7 with activists and 13 with non-activists) and 8 discussion groups (2 with activists and 6 with non-activists) between October 2019 and February 2020. Semi-structured interviews were conducted one-on-one, and each discussion group involved between 5 and 8 participants.

## Data collection and ethical considerations

A semi-structured discussion guide with open-ended questions was used to explore topics related to gender. This guide was designed after reviewing the literature and in close cooperation with the experience and knowledge of our research team and a youth advisory group. The topics addressed were: perceptions on masculinity and being a man; perceptions on femininity and being a woman; and comparison of their perceptions with those of other generations (grandparents, parents, friends. . .). These topics were raised through general questions both in the semi-structured interviews and in the discussion groups to follow an open and spontaneous conversation dynamic. Previously, 5 semi-structured interviews were conducted to verify the guide and check its adequacy and effectiveness, as well as to make any necessary modifications.

Different discussion group combinations were designed taking into account the sexual orientation, gender and involvement or not in activism criteria to enable a space where the participants could express themselves comfortably and fluently by being surrounded by people with a similar profile. Discursive inhibition that could occur due to the presence of other people in

**Table 1. Sociodemographic information of the participants.**

| Code | | Sexual orientation | Involvement in activism | Gender | No. participants |
|---|---|---|---|---|---|
| **Semi-structured interviews** | 1_W | Heterosexual | No | Woman | 1 |
| | 4_W | | | | |
| | 8_W | | | | |
| | 10_W | | | | |
| | 39_W | | | | |
| | 37_W | Bisexual | | | |
| | 41_W | | | | |
| | 32_W_ACT | Bisexual | Yes | | |
| | 34_W_ACT | | | | |
| | 7_M | Heterosexual | No | Man | |
| | 22_M | | | | |
| | 24_M | | | | |
| | 36_M | | | | |
| | 2_M | Bisexual | | | |
| | 6_M | Homosexual | | | |
| | 30_M_ACT | Heterosexual | Yes | | |
| | 31_M_ACT | | | | |
| | 21_M_ACT | Homosexual | | | |
| | 38_M_ACT | | | | |
| | 5_M_ACT | Bisexual | | | |
| **Discussion groups** | GD4 | Homosexuals and bisexuals | No | Women | 5 |
| | GD6 | Heterosexuals | | | 7 |
| | GD8 | Heterosexuals, homosexuals and bisexuals | | | 6 |
| | GD2 | Heterosexuals, homosexuals and bisexuals | Yes | | 7 |
| | GD3 | Homosexuals and bisexuals | No | Men | 5 |
| | GD5 | Heterosexuals | | | 7 |
| | GD7 | Heterosexuals, homosexuals and bisexuals | | | 8 |
| | GD1 | Heterosexuals, homosexuals and bisexuals | Yes | | 8 |

the conversation was avoided as much as possible. The final composition of each group is included in Table 1.

The qualitative fieldwork was conducted by the PositivMasc research team and each interview and discussion group lasted between 1 and 2 hours. All interviews were conducted face-to-face in Spanish (the participants' mother tongue).

In keeping ethical clearance, written informed consent from the study participants was obtained prior to enrolment. This informed consent described the study aims, how the study will be conducted, the participants, their rights, potential advantages of participating, how confidentiality will be kept and how to contact the main researcher regarding any questions/clarifications. Confidentiality was guaranteed according to the European Union General Data Protection Regulation (2016/679) and the application of the social science's deontological practices. Ethical approval was sought from the Ethical Committee of the University of Alicante in Spain (reference number UA-2019-04-15).

## Analysis

Semi-structured interviews and discussion groups were recorded digitally and transcribed verbatim to carry out a sociological analysis of the discourse system [35] using the qualitative analysis software Atlas.ti. Following Conde's approach, discourses are social productions and

practices, integrated in and in relation to a system of discourses, therefore we analyzed the narrative configurations and discursive positions that emerged in the sessions with the young people [35].

Firstly, we focused on the discussion groups by analyzing the narrative configurations that emerged when talking about masculinities and femininities, so that the essential dimensions that structured the dialogues of the participants were identified. Furthermore, the positions from which they spoke and produced their discourses were analyzed. Secondly, we reviewed the interviews, that involved more elaborate and developed discourses, something that is common from a closer conversation generated between the interviewee and the interviewer. This enabled to put into dialogue and complement the analysis of the discursive positions and narrative configurations previously identified in the discussion groups. The analytical process was led by the first author, but a continuous process of discussion was carried out with the second and third authors, who were also familiar with the fieldwork and the transcripts. They reviewed periodically the materials with the ongoing analysis produced by the first author, and they provided guidance and feedback throughout the process to ensure the coherence, rigor and validity of the analysis.

## Results

First, we describe the socially dominant discourse on gender that the young people explained at the beginning of the interviews and discussion groups, delving into the dimensions that structure their narrative configuration. Second, we highlight the main discursive positions that the participants adopted regarding the socially dominant discourse, showing the differences that were identified according to whether they were involved or not in feminist activism and their sexual orientation.

## The socially dominant gender discourse

In the interviews and discussion groups, the young people began the conversations by describing the discourse on gender that they perceive prevails in society, which establishes a social reality confronting what is masculine with what is feminine in a heteronormative space. This is a dichotomous reality where certain behaviors, qualities and rules are attributed to men and others to women, under the precept of biological sex:

> *"Through history, society has marked a series of behaviors or things that can be done and things that determine that a person is a man or a woman (. . .). It is rooted and affects everything that is not heterosexual" (GD3).*

According to what was specified, this socially dominant discourse *"has always been put into your heads by society" (GD6).* Therefore, when the participants were asked about what they understand as masculinity and femininity, they usually began to talk about former perceptions, establishing a temporary distinction between "before" and "now". On the one hand, when discussing femininity in the past, they used the metaphor of reins to highlight a subordinate, secondary and compliant role granted to women compared to men:

> *"We have always been told that being a woman is a sign of weakness, that a man has to take the reins and you have to follow" (GD4).*

In this same sense, another metaphor was used regarding slavery in relation to the lifestyle they have observed in previous generations. This implies a completely depersonalized

perception of women. They highlighted that femininity entails meeting beauty standards for men (slim physique, make-up, wearing skirts or dresses, having nice hairstyles. . .) and being relegated to care work, with no social recognition or economic remuneration:

> *"Before, being a woman was being a slave, almost. My grandmother was basically a slave. She was always getting ready and living for her man and children, to be pretty for him, and her life was always secondary" (2_M).*

On the other hand, they explained that they have been told about masculinity linked to various forms of power over women: access to the labor market, economy, enjoying leisure activities, in decision-making of any kind:

> *"Masculinity was a man who works, who goes home and dinner is ready, if he does not like the food, he tells his wife off, saying that he is going to the pub with his friends" (6_M).*

Furthermore, they explained that masculinity is like a constant performance with the same script for all men, where it was essential to seem emotionally and physically strong, be secure, brave, tough and boast about an active sex life, among others. This way of referring to masculinity reveals that young people perceive a certain artificial and imposing character in the need to publicly demonstrate certain behaviors and roles.

> *"Acting like a brave person, at certain times strong, others aggressive" (1_W).*

> *"This template of a masculine man that has no feelings, that is strong, dominant, and that thinks he is superior to others and brings others down" (38_M_ACT).*

However, when they thought about masculinities and femininities these days, they considered a more introspective perspective where they questioned if there were significant differences regarding the past. They said they have observed changes regarding women's rights, legislation and social movements towards gender equality. Nevertheless, they mentioned that a binary vision still persists of understanding masculinity and femininity based on opposition and dependence on each other:

> *"I still understand femininity and masculinity as something very binary, because if you ask what masculinity is, we will continue to think masculinity is being protective, hard, strong, that position of hierarchy and domination. And femininity as docile, submissiveness" (32_W_ACT).*

They use metaphors referring to fighting *"now we are fighting against it" (GD4)* or dragging *"we are simply dragging an old mentality" (GD4).* They warned that people, regardless of their profile (gender, age, studies, etc.), are influenced by this socially dominant discourse as different parts of their lives (culture, family, education, work. . .) are full of it.

> *Participant 1: "It has been inherited from before, to reproduce parental ways of dominance, control and so. A bit modified due to the influence of different contexts, but in the end it is reproduced".*

> *Participant 2: "Machismo is all around us in our culture and education, and in the end you are influenced by that idea" (GD7).*

Furthermore, when they considered the present, they described that the dominant ways of understanding masculinity and femininity in current society interact and are elaborated in close relation to sexual orientation. Introducing this dimension could be due to, as they explain, a greater visibility of sexual diversity in comparison to former decades. However, visibility does not mean there is no discrimination.

> *"Being less of a woman has been wrong and being a lesbian has been wrong, well they are put together. Being less of a man and being gay are wrong, together. And then you have this threshold of: if you are a very feminine woman that always wears dresses, you are a perfect heterosexual woman, society accepts you. If you are a woman that wears tracksuit, you have a problem, you are a lesbian and less of a woman. The same happens with men: if you are more sentimental, you have a problem, you are classed as gay, and as it is wrong to be gay in society, well you are digging yourself a hole. And you are classed like that"* (GD3).

This narrative configuration, where two polarized and hierarchical dimensions essentially interact (masculine-feminine and heterosexual-homosexual), establishes the foundations on which the socially dominant discourse is built (Fig 1). Masculinity is understood as the power position in a system of gender relations, and heterosexuality as a socially accepted sexual orientation. Heterosexual masculinity, of cisgender men, would hold the most privileged social position. Interactions among both dimensions are a mechanism to feed back and perpetuate the existing subordinations, violence and constraints.

Participants explained that society tends to classify the opposite gender who does not identify as heterosexual in a derogatory way. Homosexual women are criticized for being "false males". The attribute of "macho", which is positive for men, is applied to them as an insult. Homosexual men are criticized for being "feminine", which is supported by a fear and discrimination of being effeminate. Finally, bisexuals embody a vanishing point and remain invisible as an option, being at the mercy of that binary categorization. In addition, they mention that those people who transgress the roles assigned to their gender are usually categorized by society as homosexuals, with pejorative connotations.

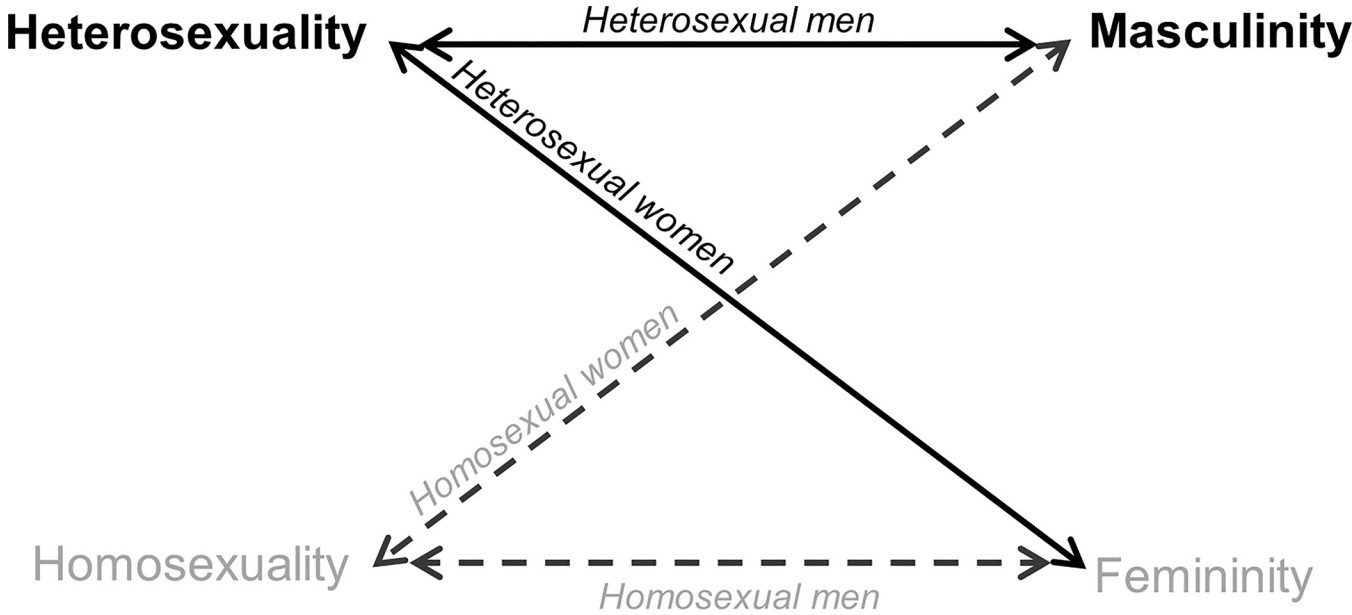

**Fig 1. Narrative configuration of the socially dominant gender discourse.**

### Discursive positions: Locations from a personal perspective

Once they explained the socially dominant discourse on masculinities and femininities, the participants began to position themselves from a personal perspective based on their socio-biographical and experiential coordinates, articulating different types of discourses. As seen in Fig 2 five discursive positions were identified: 1. Discourse on masculinity as a comfort zone, 2. Discourse on femininity as fear for retaliation, 3. Discourse on empowered femininity, 4. Discourse on deconstructed masculinity and 5. Discourse on the abolition of gender. They were structured around two axes: the adjusted-unconventional axis and an active-passive subject axis. Firstly, the adjusted-unconventional axis refers to the position they adopted regarding the socially dominant gender discourse. Closer to the "adjusted" position are the discourses in the socially dominant discourse, reproducing the traditional, binary and hierarchical gender construction. Closer to the "unconventional" position are those that transgress the socially dominant discourse, adopting an independent and reflective role regarding traditional gender mandates. Secondly, the active-passive subject axis refers to the positioning of the participants regarding possible changes towards gender equality. Closer to the "active subject" position are the discourses that are classed as responsible for current gender inequalities and are considered agents for change towards equality. Closer to the "passive subject" position are those that feel oblivious to the change towards equality, therefore avoiding responsibilities. These discursive positions, which are not mutually exclusive, are observed below and several differences were evident according to the involvement or not in feminist activism and sexual orientation.

**Discourse on masculinity as a comfort zone.** This discourse, as especially detected in the discussion groups in which heterosexual non-activists participated, revolved around the idea of masculinity as a comfort zone for men. It approached the reproduction and normalization, whether intentional or not, of the standards associated with hegemonic masculinity. The men who shared this discourse expressed conformity within the hegemonic attributes, with generally three argumentative strategies: they justified themselves by recriminating the lack of other non-hegemonic masculinity references, they supported this by means of biological claims that understand masculinity as something static and they resorted to elements such as individual idiosyncrasy or freedom of choice, which support the socially dominant discourse.

> "It is what you see in films, series, adverts, everywhere and you assume that it is what you have to do. It is not imposed, it is what we have been told to do" (GD5).

> "Men have always been like that, always. I have always said that men are like that, brave, they do not cry" (22_M).

Despite the fact that many people expressing this discourse recognized that *"it is something I had never considered" (GD5)*, they perfectly understood the behaviors and roles that they have to adopt to be considered masculine and which ones not, displaying a great repertoire when asked about it. This masculinity has permeated in such a way that they feel that they must maintain the title of "masculine", despising what is socially considered feminine and what transgresses hetero-sexuality: *"Things like 'don't cry, that is what women do' or "don't be a fag and do it'" (GD5)*.

In this sense, there are women that had the idea that masculinity (hegemonic masculinity in this case) is a comfort zone for men, bringing social pressures to which they have been subject to light. They indicated that, although men have a socially privileged position that exempts them from the structural and social subordinations that women suffer, their position involves assumptions. When men do not fulfil those assumptions, they are criticized and discriminated. Being part of that hegemonic masculinity, even in a simulated way, becomes a way to avoid possible punishments.

**SOCIALLY DOMINANT GENDER DISCOURSE**

More **adjusted** to the socially dominant gender discourse

*Discourse on masculinity as a comfort zone*

**Passive subject** to change towards gender equality

*Discourse on femininity as fear for retaliation*

*Discourse on empowered femininity*

*Discourse on deconstructed masculinity*

*Discourse on the abolition of gender*

More **unconventional** regarding the socially dominant gender discourse

**Active subject** to change towards gender equality

**Fig 2. Map of discursive positions.**

*"Men are not scared to go out like us [women]. I do think they are scared of other things, but they do not show it as they have so much social pressure. . . When there is a stereotype for women, it also affects men. For example, a man that is more sensitive cannot express himself or show his feelings because of the pressure they have to be brave"* (GD6).

In this discourse, the change towards gender equality is related to aspects like race or social class, denying any relation to the ways of understanding masculinity or possible actions take in this regard.

*"The roles and standards that you show do not affect those of others, but the other evolves as well, adapting to it, but that does not mean they [women] directly cannot do things. So, I think that it can change, but it is not directly related to not changing what we offer as men"* (GD1).

In this sense, men are presented as free of responsibility and agency in contributing to the eradication of gender inequalities. This came with a pessimist perspective that gender inequalities would be impossible to eradicate.

**Discourse on femininity as fear for retaliation.** The following discourse was mainly identified between non-activist heterosexuals. The main part of their discourse was characterized by fear. In particular, they indicated that it is an imposed fear (*"we are made to feel scared"* GD6) and feeling fear (*"in the end you don't trust"* GD6) that define femininity.

On the one hand, they talk about an imposed fear when referring to all kinds of violence (i.e., rape, sexual abuse, psychological abuse, harassment), which in turn is promoted from various spheres such as the media, pornography, the family environment, work, formal and informal education, among others. Thus, femininity was defined as a permanent constraint upon women, undermining their own personal development: experimentation with sexuality, decision-making regarding physical appearance (i.e., clothing, waxing), profession choice, and enjoying certain leisure activities. In a context of gender binarism, they explained that it corresponds to the worst side of things: the one who is blamed, the one who is ignored, the one who is made invisible, subject to the privileged side of masculinity.

*"When we talk about femininity, I think about fear and that society makes us scared for what can happen during the day, at night, in the club, in the park for being a woman. . .We have been ignored and we have been overprotected"* (GD6).

On the other hand, when they talk about feeling fear, they referred to a fear that has been imposed so they feel it in an irrational way and it affects their wellness. In the closer context of the interviews, some women reported that they have faced a fear of men in their daily lives as society has pushed them towards having a generalized view of men as potential dangers. In addition, having been socialized within the dictates of the socially dominant discourse, they declared that they are forced to develop their personal identity in accordance with the standards of traditional femininity as, otherwise, they perceive a high risk of being discriminated against by society.

*"The system is like a block where you follow cisheternormativity, or if you step outside that, that is a reason for discrimination, abuse, etc. So, when you do not fit in with those categories that are exclusionary, it seems that you are no longer valid in the system, you are no longer acceptable, you are already a freak. You have to develop your free personality always constricted within the dictates that constrain you to two things: feminine or masculine"* (10_W).

Another aspect they highlighted is that, although theoretically women have legislation that improves their rights, in practice conditions have worsened. One of the examples they mentioned is women being included in the labor market while care work is still imposed on them as in the past. It reveals the double burden to which they are currently subjected:

*"What happens is that women have entered the labor market but are still subject to the same constraints as before at home, combining this with having to be a successful employee. It is like a double-edged sword" (GD1).*

When the young people discussed a change towards gender equality, they treated it as something that depends on new generations. From an optimistic and relatively unrelated prism, they gave young cohorts the responsibility of shaping new ideas on femininity so that a generational change gradually takes place as the cohorts are renewed.

**Discourse on empowered femininity.** This discourse, which is mainly present in heterosexual women activists and homosexual and bisexual women involved or not in activism, was distinguished from others by expressing opposition to the current gender system. They understood femininity as a construction that should be subjective and individual by people who identify themselves as women, whether trans or cisgender. This does not necessarily imply not having all the elements of the social construction of traditional femininity, but it does imply its questioning in order to get rid of all the constraints that surround it. In other words, they defended a self-definition based on plurality and personal freedom.

*"For me, femininity is simply the fact that you feel feminine by being a woman. For example, if you feel that being a woman is wearing a tracksuit, trainers and having short hair, as is my case, that is feminine. Being feminine is not just me wearing a pink skirt, with tights or long combed hair" (37_W).*

Based on a hierarchical scenario and a position of subordination, they described femininity as an engine of empowerment and as a source of energy to deconstruct hegemonic masculinity. In terms of this empowerment, they criticized the attributes that they consider to be socially feminine by claiming the need to actively transgress them. The women who have gone through this process of transgression say that they face discriminatory experiences: when playing sports, in their professions that were traditionally associated with men or having a dress code that perpetuates the objectification of women, in the home itself, in relationship with another woman, among others. Establishing resistance and negotiations in these circumstances are considered to be key.

*"It is not feminine to play sports, having an initiative regarding a relationship, considering what you want and what you do not want, because we supposedly always leave it to men. There are boys that, when you break up with them, they get violent and it messes with their mentality. We are there to fight that with men and change their minds. We are also here to talk to women who are not so close to feminism" (GD2).*

People with these discourses feel that the process towards gender equality is the responsibility of their generation. They are classed as possible agents of change in a time of historical change.

*Participant 1: "I think we are the generation to create change".*

*Participant 2: "Yes, we are part of the process I think".*

*Participant 3: "The thing is we are now going through that change. Former generations are like: 'I don't know what to do', so in that sense we also have to help them" (GD6).*

Participants speak of a two-fold change. Firstly, and internal change regarding the way of understanding femininity from cooperation and diversity. Secondly, external change by accompanying other generations in this process of transformation towards equality.

**Discourse on deconstructed masculinity.** This discourse can be observed especially among activists of all sexual orientations (heterosexuals, homosexuals and bisexuals) and homosexual and bisexual non-activists. This discourse is characterized by deconstructing hegemonic masculinity. They displayed a deeply reflective narrative, especially in the interviews, which introspectively delved into the individual, social and structural implications that hegemonic masculinity has for both men and women. They condemned the fact that there is a *"lack of self-reflection" (7_M)* by men, alongside the need to break the normalization of machismo. For instance, in the following quote, a participant states that the feminist search for a plural and equal masculinity among those self-identified as men (trans and cisgender) is crossed by cisheteropatriarchy, capitalism and racism; axes which are considered to be essential in order to subvert the current reality:

*"I think it is essential to achieve this deconstruction and open our eyes to pay attention to these situations [machismo] that seem to unfortunately be normalized, to undo them, question them and challenge them. Men need to report this, go beyond it and say: 'this is not OK'. That is where the new masculinities emerge for me. For me, new masculinities are a masculine identity stripped of that toxicity and constraints, they are much freer. Anything that goes beyond the dictates of a white, cisgender, hetero man. This comes from cisheteropatriarchy, at the hand of capitalism and racism" (30_M_ACT).*

The men recognized that this is not a simple task, as the first stone set in the path is social pressure by peers. Showing a vulnerable and sentimental side, and stepping aside from the strength that hegemonic masculinity establishes, becomes a difficult task due to the fear of being socially judged and being categorized as "feminine" or "homosexual" in a negative sense, with the discrimination that this carries.

*"I think certain behaviors that are not so focused on strength are difficult to show. They are not impossible to show, but I do think that it is more embarrassing and that there is greater fear. We [men] do not feel safe in certain contexts with people with whom you may not trust, with certain friends or in groups of work relationships. But when I have been among an activist circle it is easier and I have felt much more comfortable when speaking, when analyzing my own behaviors, because it is rewarding. I have seen that my context influences a lot" (7_M).*

They considered it essential to join forces to create safe environments where masculinity can be expressed freely. They indicated that all men embody hegemonic masculinity, consciously or unconsciously, and they are responsible for detaching themselves from the privileges that perpetuate the oppression of women. In this sense, they mention the need to continue moving forward in educating about feminism and promoting a greater activist movement that encourages dialogue and masculine reflexivity. This should be carried out with the support of institutions.

**Discourse on the abolition of gender.** The following discourse is mainly observed with activists of all sexual orientations (heterosexuals, homosexuals and bisexuals). In the same

sense as deconstructed masculinity and empowered femininity discourses, this discourse has a critical approach focused on deconstructing the socially dominant discourse, but in this case it was conducted with the aim of abolishing gender. They describe the current gender system as a "Judeo-Christian tradition" which has had an economic benefit for the capitalist system. As they explained, this has meant a condemnation of heterogeneity that characterizes humanity, exerting violence on other identities such as transgender or other sexual orientations like bisexuality.

*"I think it is essential to understand that we come from a western philosophy that as always reduced reality to dichotomies, which goes hand in hand with a Judeo-Christian tradition that has cultural baggage and unfortunately still has a lot of weight. So, reality has been reduced to two genders and you cannot choose it when it is society that gives you it. You do not choose the clothes you wear, the patriarchy chooses it hand in hand with capitalism. (. . .) The reality, which is so diverse and so widespread, is enslaving and confining it"* (32_W_ACT).

In order to abolish gender, they admitted that it is necessary to go through a long process to work on current inequalities. Once this has been achieved, the concept of gender will have to be let go:

*"One of my goals as an activist is to abolish gender. This is something I do not want to do straight away as this will not result in anything positive. If you say tomorrow that there is no gender, the only thing you achieve is making violence against women invisible and many other things become invisible. But as an idea, I think it is marvelous. It is the point we should reach, not just have legislation to get rid of it, but as a social idea, the day it is gone, that day will be better for humanity"* (38_M_ACT).

However, people with this discourse recognize that abolishing gender requires an invaluable period of time due to the innumerable changes and transformations that must be carried out in the different spheres of society such as the educational system, the media or the legislative system. Some participants even go so far as to consider that achieving a genderless society is a utopia.

## Discussion

The sociological analysis of the discourse system in this study provided the opportunity to delve into the bases that structure the socially dominant gender discourse and how young people discursively position themselves regarding it. Our findings show that this traditional and dominant discourse is perceived to practically be unaltered despite years going by and it is present in different fields (social, economic, political, etc.). Femininity is perceived as subject to the mandates of masculinity. Moreover, it is not only constructed according to hierarchy between masculine and feminine, but also intertwining unequal relations between heterosexuals and homosexuals. This study has also identified differences in the way in which young people position themselves regarding said discourse, according to their sexual orientation and involvement in feminist activism. Discursive positions that reproduced the socially dominant discourse were especially detected among non-activist heterosexuals. Conversely, the transformative discursive positions involved in searching for gender equality were present mainly in activists of all sexual orientations (heterosexual, homosexual and bisexual) and non-activist homosexuals and bisexuals. This suggests that there are subcultural spaces independent from the socially dominant discourse, but that do not result in a systemic social change.

Our findings reveal that the participants of this study still construct femininities and masculinities in a continuous fight in the face of oppression established by the socially dominant discourse, where the hegemonic masculinity reigns. In line with Marcos-Marcos et al., hegemonic masculinities must be understood as a process and not just as a set of attributions [36]. We have verified in the analyzed discourses that, in said process, gender intersects with sexual orientation [37]. Therefore, and in line with other research [8, 9], we observe that masculinity and heterosexuality are considered to be socially superior, and femininity and homosexuality are considered to be inferior. Homosexual people are socially categorized with a gender other than their own, with a damaging background [38]. It has been confirmed that bisexuality, by not falling within these dichotomies, represents a vanishing point that makes people who identify as such invisible and discriminated [39], forcing them to fit into the binary scheme of sex, gender and sexuality.

The experiences stated by the young participants show that homophobia and gender vigilance continue to exist, although adopting more subtle ways [40]. Bridges and Pascoe already expressed the importance of understanding homophobia as a gender practice [41]. The young men stated that they continuously struggle to avoid being linked to homosexuality, as becoming a "fag" has much to do with failing at masculinity or revealing femininity. This is what Pascoe named the "fag discourse", understood as a disciplinary discourse through which boys condition themselves, and each other, under the guise of joking interactions [42]. "Fag" is not only associated with homosexual men, but it is an identity in which any man of any sexual orientation can be temporarily classified. Consequently, this involves a potent disciplinary mechanism that controls the way in which masculinity, and femininity collaterally, is constructed in young people for a fear of the possible punishments and social discrimination that occupying the "fag" position implies.

Several discursive positions by the young people were established as a socially dominant gender discourse. One of them was the masculinity discourse as a comfort zone, showing the "fitting in" pressure around the idea of hegemonic masculinity that men experience since they are young. Another discourse was femininity as fear for retaliation, which reflects the subordination that women experience every day. Both are adapted to and comply with the bases of hegemonic masculinity. The notion of "accountability" [43, 44] could be key to interpreting them. It refers to the anticipated assessment of people of their behaviors, from which they can adapt their actions depending on how they can be interpreted by others in the social context where they occur. The participants try to identify as male or female through interactions between sex, gender and sexuality. As part of this socially structured interaction and in order to avoid being negatively judged, they show their masculinity and femininity through dominant discourses and practices that are interpreted as such.

In accordance with what has been described in previous research regarding the masculinist discourse [19], the background of the discourse on masculinity as a comfort zone is based on biological arguments, or they resort to justifications such as individual idiosyncrasy or freedom of expression, which detaches masculinity from its socio-historical production and hides the structural implications when it comes to understanding current gender relations. The discourse on femininity as a fear of retaliation represents a clear example of the "emphasized femininity" that Connell coined [5]. There is a clear scope of action regarding these discursive findings, as understanding how dominant hegemony is configured can help to propose future strategies to disarticulate and delegitimize it [6].

In this research, people whose discourses show a rebellious and transgressive relationship with the current gender order are mainly activists of all sexual orientations and non-activist homosexuals and bisexuals. Nonetheless, these encountered discourses seem to be a tactical proposal, so we should continue to research the elements that can restrain gender equality and

the strategic and political proposals that are demanded. Firstly, in the discourse on deconstructed masculinity, they advocate "inclusive masculinities" as the true challenge to hegemonic masculinity [45], understood as the proliferation and social inclusion of multiple masculinities with less hierarchy. However, the young people state their surroundings are insecure to carry them out in real practice, lacking educational and social policies that favor personal and structural changes in gender equality. Secondly, the discourse on empowered femininity is similar to what Howson classed as "ambivalent femininities", as strategic combinations of resistance and cooperation are adopted, the critique and embrace of femininities, but the way of protest that it promotes only involves idiosyncratic and conjunctural tactics [6]. Finally, in the discourse on gender abolition, an intellectual leadership is begun to be developed, establishing a progressive and complex project, sometimes seen as utopian, in order to promote complete independence in gender construction and its corresponding inequalities. Thus, it is necessary to delve into the meeting points of feminist activism with LGBT activism in future studies, as well as how they may be in a unique position in the progress towards gender equality.

This study includes some limitations that should be considered. Although we researched how sexual orientation and involvement in feminist activism may influence the discourses around femininity and masculinity, the sample could also be increased by including other gender identities (such as transgender people) and sexual orientations (such as asexual people) [e.g., see 46], geographical locations and age groups to a broader understanding of the study subject. As mentioned in the methods section, we focused on certain categories so as not to saturate the analytical combinations and to ensure a sample with structural representativeness in the established categories. We also did not examine how other variables may intersect, hence future research could investigate other variables as class, ethnicity or religion. Additionally, social desirability bias is a risk that can occur in any study. The topic addressed in this study can be quite personal and uncomfortable for participants that favor social desirability in their discourses. To reduce this bias, discussion groups were designed according to sexual orientation, gender and involvement or not in feminist activism so participants would feel more inclined to share information with others like themselves. Before each session, the interviewers encouraged the participants to express themselves freely and to act in a non-judgmental manner. In addition, the guide that was used in the sessions intended to encourage a dynamic, open and spontaneous conversation to achieve a better understanding of the issues raised during the sessions and to freely encourage discussion.

## Conclusion

The socially dominant gender discourse, in which hegemonic masculinity and heteronormativity prevail, continues to consciously and unconsciously permeate the way in which young people understand and construct their idea of masculinity and femininity, establishing a whole system of inequalities and power imbalances. This study has intended to contribute to the understanding of the configuration of inequalities in gender relations. It empirically adds to the current knowledge of the interplay of masculinities and femininities and how interactions with sexuality take place. Although discourses that transgress and search for independence from the traditional gender construction are observed, mainly in activists of all sexual orientations and homosexual and bisexual non-activists, they are restricted by the normative structure that said socially dominant discourse unfolds. This study has revealed the relevance of including discourses of heterosexual and LGB youth, in addition to the discourses of people involved or not in feminist activism. Our findings indicate that there is a need to continue working on a gender-transformative approach [47–49] through interventions from an early age. They

should address aspects that became known in this research, such as the lack of non-hegemonic and egalitarian masculinities references, psychological and emotional repression of men, social peer pressure, homophobia and biphobia, or the deconstruction of biological beliefs about gender roles to promote gender equality and social justice.

## Acknowledgments

The authors gratefully acknowledge the contributions of the young people who participated in this research.

## Author Contributions

**Conceptualization:** Ariadna Cerdán-Torregrosa, Daniel La Parra-Casado, Carmen Vives-Cases.

**Data curation:** Ariadna Cerdán-Torregrosa.

**Formal analysis:** Ariadna Cerdán-Torregrosa.

**Funding acquisition:** Carmen Vives-Cases.

**Investigation:** Ariadna Cerdán-Torregrosa, Daniel La Parra-Casado, Carmen Vives-Cases.

**Methodology:** Ariadna Cerdán-Torregrosa, Daniel La Parra-Casado, Carmen Vives-Cases.

**Project administration:** Carmen Vives-Cases.

**Supervision:** Daniel La Parra-Casado, Carmen Vives-Cases.

**Validation:** Daniel La Parra-Casado, Carmen Vives-Cases.

**Writing – original draft:** Ariadna Cerdán-Torregrosa.

**Writing – review & editing:** Ariadna Cerdán-Torregrosa, Daniel La Parra-Casado, Carmen Vives-Cases.

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
