## [Decision Letter · Decision Letter 0]

5 Apr 2023

PONE-D-22-30849“It is what we have been told to do”: Masculinities and femininities crossing with sexual orientation and feminist activism in SpainPLOS ONE

Dear Dr. Cerdán-Torregrosa

Thank you for submitting your manuscript to PLOS ONE. After careful consideration, we feel that it has merit but does not fully meet PLOS ONE’s publication criteria as it currently stands. Therefore, we invite you to submit a revised version of the manuscript that addresses the points raised during the review process.

I have made a few comments below that most likely will only require a brief bit of your time.  the reviewers did not make any suggestions for revisions.

Please submit your revised manuscript by April 21, 2023 If you will need more time than this to complete your revisions, please reply to this message or contact the journal office at plosone@plos.org. Please include the following items when submitting your revised manuscript:A rebuttal letter that responds to each point raised by the academic editor and reviewer(s). You should upload this letter as a separate file labeled 'Response to Reviewers'.A marked-up copy of your manuscript that highlights changes made to the original version. You should upload this as a separate file labeled 'Revised Manuscript with Track Changes'.An unmarked version of your revised paper without tracked changes. You should upload this as a separate file labeled 'Manuscript'.If applicable, we recommend that you deposit your laboratory protocols in protocols.io to enhance the reproducibility of your results. Protocols.io assigns your protocol its own identifier (DOI) so that it can be cited independently in the future. For instructions see: https://journals.plos.org/plosone/s/submission-guidelines#loc-laboratory-protocols. Additionally, PLOS ONE offers an option for publishing peer-reviewed Lab Protocol articles, which describe protocols hosted on protocols.io. Read more information on sharing protocols at https://plos.org/protocols?utm_medium=editorial-email&utm_source=authorletters&utm_campaign=protocols.

We look forward to receiving your revised manuscript.

Kind regards,

Mary Diane Clark, PhD

Academic Editor

PLOS ONE

Journal Requirements:

“This work was supported by GENDER NET Plus Co-Fund (Reference 2018-00968) and the Ministry of Science and Innovation of Spain (Reference PCI2019-103580) as part of the PositivMasc project from which CVC is the principal investigator. This study has also been conducted within the predoctoral grant received from the Ministry of Universities of Spain (Reference FPU19/00905) from which ACT is recipient.”

Additional Editor Comments (if provided):

thank you for. your manuscript. The reviewers made no suggestions but I have a few minor changes I would like to see. The topic is interesting and the work can be transformative.

Line 413 Can you have a better conclusion to this section? It needs to have a summary rather then ending with a quote

Line 420 can you delete the … after harassment and start the ( ) with i.e.,

Also same on line 425

And the etc at the end of line 425

Line 427 same three periods please add. Alternative punctuation

Line 426 ---gender binary. ---- should this be binary gender?

Line 496---again you need a concluding paragraph

Line 562---again need a concluding paragraph

Reviewers' comments:

Reviewer's Responses to Questions

**Comments to the Author**

1. Is the manuscript technically sound, and do the data support the conclusions?

Reviewer #1: Yes

Reviewer #2: Yes

2. Has the statistical analysis been performed appropriately and rigorously? 

Reviewer #1: N/A

Reviewer #2: N/A

3. Have the authors made all data underlying the findings in their manuscript fully available?

Reviewer #1: No

Reviewer #2: Yes

4. Is the manuscript presented in an intelligible fashion and written in standard English?

Reviewer #1: Yes

Reviewer #2: Yes

5. Review Comments to the Author

Reviewer #1: I enjoy reading this manuscript and it helped me to see why bisexuality is a vanishing point, because it does not fit the heteronormative dichotomy. However, I would recommend you to change this label LGB to LGBT as some transgender do internalize heteronormative tendencies.

Reviewer #2: The submission "It is what we have been told to do": Masculinities and femininities crossing with sexual orientation and feminist activism in Spain is technically sound in its qualitative context using semi-structured interviews and discussion groups to identify a type of discourse that emerges in masculinities and femininities relative to sexual orientation and feminist activism in Spain. The data uncovered from the discussion groups supports the identification of a socially dominant gender discourse with the masculine approach being perceived as dominant over femininity. There also is a prevalence of reproducing that socially dominant gender discourse within non-activist heterosexuals, and the data are consistent with the understanding that there are others who try to deconstruct that gender discourse who are activists across all sexual orientations.

In terms of statistical analysis - there is no quantitative analysis presented - which is appropriate given the qualitative context of the study. Qualitative analysis was used via Atlas.ti (qualitative analysis platform) - where the authors analyzed the "narrative configurations and discursive positions" of the participants. The first author led the analytic process with the remaining authors, engaging in a continuous discussion process where guidance and feedback was provided to ensure consistency within the rigor and validity of the sociological analysis.

The results from the analysis are consistent with the identification of a socially dominant gender discourse and are explained clearly by group (seven data groups in all) using paraphrases from interviews as forms of evidence to support the identification of that gender discourse.

Finally, future direction is provided in identifying the need to continue working on a "gender-transformative approach through interventions from an early age as well as deconstructing the biological beliefs about gender roles to amplify and promote gender equality/social justice."

6. PLOS authors have the option to publish the peer review history of their article (what does this mean?). If published, this will include your full peer review and any attached files.

Reviewer #1: No

Reviewer #2: No

---

## [Author Response · Author response to Decision Letter 0]

19 Apr 2023

PONE-D-22-30849

“It is what we have been told to do”: Masculinities and femininities crossing with sexual orientation and feminist activism in Spain

Dear Academic Editor Dr. Mary Diane Clark, 

We thank you for the opportunity to submit the manuscript revised. We also would like to thank you and the reviewers for the valuable comments. Below we provide detailed responses on how we addressed each point raised during the review process. We have submitted a file labeled 'Revised Manuscript with Track Changes' with changes marked using red colored text, and a file labeled 'Manuscript' without tracked changes. We hope that these revisions will serve to meet PLOS ONE’s criteria. 

Looking forward to hearing from you soon.

Kind regards,

Ariadna Cerdán-Torregrosa

Response to Academic Editor comments:

Journal Requirements:

- We have reviewed the documents listed and the other information available on the journal's website to ensure that our manuscript meets PLOS ONE's style requirements. A few changes have been made accordingly. Firstly, we have added full stops to the titles of Fig 1-3 and Table 1. Secondly, the naming of the figure files have been changed to "Fig 1.tif" and "Fig 2.tif" and we have made sure that our files comply with PLOS ONE's format.

“This work was supported by GENDER NET Plus Co-Fund (Reference 2018-00968) and the Ministry of Science and Innovation of Spain (Reference PCI2019-103580) as part of the PositivMasc project from which CVC is the principal investigator. This study has also been conducted within the predoctoral grant received from the Ministry of Universities of Spain (Reference FPU19/00905) from which ACT is recipient.”

- We have amended the Funding Statement following your recommendations. The Funding Statement now is as follows:

“This work was supported by GENDER NET Plus Co-Fund (Reference 2018-00968) and the Ministry of Science and Innovation of Spain (Reference PCI2019-103580) as part of the PositivMasc project from which Carmen Vives-Cases is the principal investigator. This study was also conducted within the predoctoral grant received from the Ministry of Universities of Spain (Reference FPU19/00905) from which Ariadna Cerdán-Torregrosa is recipient. The publication costs were supported by AICO, Generalitat Valenciana (2022-2024) (Reference CIAICO/2021/019). The funders had no role in study design, data collection and analysis, decision to publish, or preparation of the manuscript. There was no additional external funding received for this study.”

- We have checked our cites and reference list to ensure that it is complete and correct and have detected the following error which has now been fixed: on line 585 the citation "[36]" has been changed to "[37]". We have not cited papers that have been retracted.

Additional Editor Comments (if provided):

thank you for. your manuscript. The reviewers made no suggestions but I have a few minor changes I would like to see. The topic is interesting and the work can be transformative.

- We thank you for your enriching review of our manuscript and for the important comments you have made. Below we provide detailed answers for each of these comments, explaining how we have made changes in the manuscript based on them.

Line 413 Can you have a better conclusion to this section? It needs to have a summary rather then ending with a quote

- Thank you for your suggestion. We have added the following paragraph between lines 414 and 416: 

“In this sense, men are presented as free of responsibility and agency in contributing to the eradication of gender inequalities. This came with a pessimist perspective that gender inequalities would be impossible to eradicate.”

Line 420 can you delete the … after harassment and start the ( ) with i.e.,

- Yes, we have modified the sentence in the light of your proposal. Please, see line 423:

“(i.e., rape, sexual abuse, psychological abuse, harassment)”

Also same on line 425

And the etc at the end of line 425

- We have changed the sentence, please see lines 427-428:

“(i.e., clothing, waxing), profession choice, and enjoying certain leisure activities.”

Line 427 same three periods please add. Alternative punctuation

- Agree. We have added alternative punctuation, please see line 430:

“who is blamed, the one who is ignored, the one who is made invisible, subject to the”

Line 426 ---gender binary. ---- should this be binary gender?

- We have changed the term to "gender binarism" because participants refer to the social and cultural construct based on the belief that society has only two genders: feminine and masculine, as set out in the results. We hope that this change will make it clearer. Please, see line 429:

“of gender binarism, they explained that it corresponds to the worst side of things: the one”

Line 496---again you need a concluding paragraph

- Agree. We have written a concluding paragraph, please see lines 498-501: 

“Participants speak of a two-fold change. Firstly, and internal change regarding the way of understanding femininity from cooperation and diversity. Secondly, external change by accompanying other generations in this process of transformation towards equality.”

Line 562---again need a concluding paragraph

- Yes. We have written a concluding paragraph accordingly, please see lines 568-572:

“However, people with this discourse recognize that abolishing gender requires an invaluable period of time due to the innumerable changes and transformations that must be carried out in the different spheres of society such as the educational system, the media or the legislative system. Some participants even go so far as to consider that achieving a genderless society is a utopia.”

Response to Reviewer 1 comments:

Reviewer #1: I enjoy reading this manuscript and it helped me to see why bisexuality is a vanishing point, because it does not fit the heteronormative dichotomy. However, I would recommend you to change this label LGB to LGBT as some transgender do internalize heteronormative tendencies.

- Thank you very much for the review and your positive feedback. In accordance with your recommendation, we have modified the sentence in line 657: 

“with LGBT activism in future studies, as well as how they may be in a unique position in”

However, we have kept the label "LGB" in line 690 because on that occasion we only referred to our sample, in which no transgender people participated.

Response to Reviewer 2 comments:

Reviewer #2: The submission "It is what we have been told to do": Masculinities and femininities crossing with sexual orientation and feminist activism in Spain is technically sound in its qualitative context using semi-structured interviews and discussion groups to identify a type of discourse that emerges in masculinities and femininities relative to sexual orientation and feminist activism in Spain. The data uncovered from the discussion groups supports the identification of a socially dominant gender discourse with the masculine approach being perceived as dominant over femininity. There also is a prevalence of reproducing that socially dominant gender discourse within non-activist heterosexuals, and the data are consistent with the understanding that there are others who try to deconstruct that gender discourse who are activists across all sexual orientations.

In terms of statistical analysis - there is no quantitative analysis presented - which is appropriate given the qualitative context of the study. Qualitative analysis was used via Atlas.ti (qualitative analysis platform) - where the authors analyzed the "narrative configurations and discursive positions" of the participants. The first author led the analytic process with the remaining authors, engaging in a continuous discussion process where guidance and feedback was provided to ensure consistency within the rigor and validity of the sociological analysis.

The results from the analysis are consistent with the identification of a socially dominant gender discourse and are explained clearly by group (seven data groups in all) using paraphrases from interviews as forms of evidence to support the identification of that gender discourse.

Finally, future direction is provided in identifying the need to continue working on a "gender-transformative approach through interventions from an early age as well as deconstructing the biological beliefs about gender roles to amplify and promote gender equality/social justice."

- We really appreciate the time you have dedicated to our manuscript. Thank you very much for the review and your valuable feedback.

---

## [Editor Report · Decision Letter 1]

26 Apr 2023

“It is what we have been told to do”: Masculinities and femininities crossing with sexual orientation and feminist activism in Spain

PONE-D-22-30849R1

Dear Dr. Cerdan-Torregrosa

We’re pleased to inform you that your manuscript has been judged scientifically suitable for publication and will be formally accepted for publication once it meets all outstanding technical requirements.

Kind regards,

Mary Diane Clark, PhD

Academic Editor

PLOS ONE

Additional Editor Comments (optional):

Thank you for the care you took with the suggestions from me and the reviewers. The paper is interesting while maybe depressing as these gender roles are so stubborn and resistant to equalitarian roles.

Nice paper and strong contribution to the literature.

Reviewers' comments:

none

---

## [Editor Report · Acceptance letter]

2 May 2023

PONE-D-22-30849R1 

“It is what we have been told to do”: Masculinities and femininities crossing with sexual orientation and feminist activism in Spain 

Dear Dr. Cerdán-Torregrosa:

I'm pleased to inform you that your manuscript has been deemed suitable for publication in PLOS ONE. Congratulations! Your manuscript is now with our production department. 

Kind regards, 

on behalf of

Dr. Mary Diane Clark 

Academic Editor

PLOS ONE